# The Perspective and Challenge of Nanomaterials in Oil and Gas Wastewater Treatment

**DOI:** 10.3390/molecules26133945

**Published:** 2021-06-28

**Authors:** Xiaoying Liu, Wenlin Ruan, Wei Wang, Xianming Zhang, Yunqi Liu, Jingcheng Liu

**Affiliations:** 1Collage of Environment and Resources, Chongqing Technology and Business University, Chongqing 400047, China; lxy_ctbu@163.com (X.L.); ruuanwl@163.com (W.R.); 2Key Laboratory of Complex Oil and Gas Field Exploration and Development of Chongqing Municipality, Chongqing University of Science & Technology, Chongqing 401331, China; ww530010406@163.com; 3Engineering Research Center for Waste Oil Recovery Technology and Equipment, Ministry of Education, Chongqing Technology and Business University, Chongqing 400067, China; 4State Key Laboratory of Heavy Oil Processing, China University of Petroleum (East China), Qingdao 266580, China; liuyq@upc.edu.cn

**Keywords:** oil and gas wastewater, photocatalysis technology, membrane separation technology

## Abstract

Oil and gas wastewater refers to the waste stream produced in special production activities such as drilling and fracturing. This kind of wastewater has the following characteristics: high salinity, high chromaticity, toxic and harmful substances, poor biodegradability, and a difficulty to treat. Interestingly, nanomaterials show great potential in water treatment technology because of their small size, large surface area, and high surface energy. When nanotechnology is combined with membrane treatment materials, nanofiber membranes with a controllable pore size and high porosity can be prepared, which provides more possibilities for oil–water separation. In this review, the important applications of nanomaterials in wastewater treatment, including membrane separation technology and photocatalysis technology, are summarized. Membrane separation technology is mainly manifested in ultrafiltration (UF), nanofiltration (NF), and reverse osmosis (RO). It also focuses on the application of semiconductor photocatalysis technology induced by TiO_2_ in the degradation of oil and gas wastewater. Finally, the development trends of nanomaterials in oil and gas wastewater treatment are prospected.

## 1. Introduction

Recently, despite the sharp drop in oil prices, the importance of oil as a national strategic resource is still self-evident. Shale gas, as a kind of green and clean energy strongly encouraged by the state, has made important breakthroughs in exploration and development, although there is still a certain gap in various aspects of technology compared with foreign countries at the present stage, which has aroused widespread concern [1,2]. According to statistics, the output of shale gas in China in 2018 was 108.81 × 10^8^ m^3^, an increase in 21.0% over the same period last year. By the end of 2018, a total of 1027 oil and gas fields have been discovered in China, of which the cumulative output of oil reached 69.52 × 10^8^ t, and the cumulative production of natural gas reached 2070 × 10^8^ m^3^. However, behind the vigorous development of oil and gas resources, it also causes a series of environmental problems. In the production operation activities, as most of the reservoirs are deeply buried and the geological conditions are complex, it is usually necessary to carry out activities such as water injection and fracturing. The resulting large amount of waste liquid cannot be completely used for reinjection, and a considerable part of it can only be discharged. Oil and gas production wastewater contains many complex organic components [3] (such as polycyclic aromatic hydrocarbons, aromatic hydrocarbons, oils, etc.), heavy metals [4], and natural radionuclide (NORMs) [5]. It was reported [4] that each unconventional shale gas well produces about 40 × 10^8^ m^3^ of wastewater each year. If the wastewater is discharged directly without treatment, it will pose a serious threat to the environment. Toxic chemicals are pumped into the land and pollute the air and water, bringing harm to wildlife and agricultural production, and affecting human health [6,7]. In addition, the huge water consumption of a single well in the process of drilling and fracturing will cause a certain shortage of local water supply in areas with poor water resources [8]. The up-to-standard discharge and sustainable recycling of oil and gas wastewater has become a difficult problem for the oil industry. At present, the treatment methods of oil and gas wastewater mainly include gravity separation, chemical treatment, biodegradation, and other technologies, but most of them have the problems of secondary pollution and a high economic cost.

The treatment of oil and gas wastewater by membrane separation technology can avoid demulsification, so there is no need to add any chemical reagents, which have a certain separation effect on heavy metals in wastewater, and a high removal efficiency of COD. Compared with ordinary filter media, membrane separation has a smaller pore size and a narrower pore size distribution. According to the order of membrane pore size from large to small, it can be divided into microfiltration (MF), UF, NF and RO, as shown in Table 1.

The nanofiltration membrane and the reverse osmosis membrane can operate normally under high pressure; the main difference between them lies in selectivity [9]. The nanofiltration membrane allows divalent and multivalent ions as well as some Na^+^ and Cl^−^ to pass through, while the reverse osmosis membrane refuses all ions to pass through, which can achieve the purpose of seawater desalination. As a result, these membranes are more susceptible to contamination during operation. Abdollahzadeh et al. [10] used a membrane bioreactor (MBR) with moderate halophilic bacteria to treat laboratory-synthesized high-salinity gas field wastewater. The results showed that the water turbidity was less than 2 NTU and that there was no obvious membrane pollution.

Nano-materials usually refer to ultra-fine materials whose size is less than 100 nm. They show unique properties in light, heat, electricity, magnetism, mechanics and chemistry because of their large specific surface area and high surface energy. These materials have two potential applications in wastewater treatment: (1) integrating with membrane separation technology to treat oil and gas wastewater, including NF, UF, RO separation membrane; and (2) using the photocatalytic properties of some nanoparticles (TiO_2_, ZnO_2_, Fe_2_O_3_, etc.) to realize the oxidative decomposition of pollutants under light, so as to achieve the purpose of wastewater purification [11]. This review mainly addresses the application of nanomaterials in the treatment of oil and gas wastewater, including membrane separation technology and photocatalysis technology, and summarizes the progress of photocatalysis treatment of different types of oil and gas produced by water with TiO_2_ as the research object, including its influencing factors, the main limitations faced by TiO_2_, the enhancement of photocatalytic performance, and the prospect of the future.

## 2. Modification of Membranes with Nanomaterials

Polymer is a kind of membrane material widely used in wastewater treatment, though it is limited in the application of polyvinylidene fluoride (PVDF) and polysulfone (PSF) ultrafiltration [12] due to the poor hydrophilicity and serious membrane fouling [13]. However, the pollution in the use process will lead to the reduction of separation efficiency and the increase in the recovery cost. Therefore, nanoparticles are usually used to modify the corresponding membrane materials, and, by establishing super-wetting surfaces (super hydrophobic/super oil-wet or super hydrophilic/super oleophobic), it can efficiently separate the oil and water mixture and improve the anti-fouling performance [14]. In Table 2, a number of studies on nano-modification of membranes are summarized.

Zhang et al. [20] modified PVDF membranes with titanium dioxide functional nanoparticles to prepare low-cost and reusable self-assembled monolayers. These TiO_2_ particles can improve the membrane performance by enhancing the membrane’s permeability flux and salt rejection. Because they were inorganic oxidized nanoparticles, they can also effectively enhance the chemical and mechanical stability of the membrane. Cui et al. [21] have synthesized a super-hydrophilic/underwater super-hydrophobic nanocomposite film. In other words, the PVDF membrane was modified by a polydopamine composite of nano-silica by an inverse method. The PVDF@PDA@SiO_2_ nanocomposite membrane was not only simple to prepare but could also maintain a high regeneration rate after several cycles, while the effluent emulsion TOC was lower than 30 ppm. As shown in Figure 1, its special surface micro-nano structure and pore induced capillarity phenomenon are presented on the surface after modification. This kind of composite membrane had great potential in industrial application because of its good oil–water separation, reproducibility, and versatility. Chen et al. [22] prepared nanofiltration membranes with high water flux. They prepared a sandwich-like nanofiltration membrane of PA6@GO@PA6 and intercalated hydrophilic nanoparticles such as titanium dioxide, silicon dioxide and trisilicon tetroxide with substrate (graphene oxide layered material) by means of electronic spray and electrospinning, and then assembled it layer-by-layer with polyamide-6 (PA6). The experimental XRD diagram showed that, with the addition of nano-TiO_2_ (NP), the layer spacing of the films increased, the water flux increased significantly, and the rejection rate of organic dyes remained high. This method can be used as a reference for the preparation of organic–inorganic composite membrane materials. Song et al. [23] prepared the nano-TiO_2_ nanofiltration membrane by means of a new molecular layer deposition method (MLD). The composition, thickness, and pore size of the membrane were expected to be controllable. Compared with the traditional sol–gel method, the composite membrane had more obvious advantages, including a good separation performance and antifouling performance.

Most of the membranes are made up of organic polymers. In recent years, inorganic ceramic membranes have also been widely used. Weschenfelder et al. [24] aimed at the treatment of organic matter in the produced water of an oil field in Brazil, whereby a multi-tube nano-ceramic membrane (ZrO_2_) was used to treat it. The results showed that the osmotic flow, which can meet the higher reuse requirements of offshore oil drilling platforms, can be produced under 1.5 bar, and that the material can be restored to its original permeability through the cleaning scheme of acid washing and alkali washing. It was also detected that the main pollutant accumulated was iron oxide. Considering that the composition of produced water in oil field varied greatly from well to well, the applicability of this kind of ceramic membrane needed to be further studied. Weschenfelder et al. [25] conducted a pilot-scale evaluation of a ceramic membrane (ZrO_2_) and estimated the cost according to the results of different membrane cleaning cycles. The results showed that membrane rinsing every 100 h can reduce the cost of the whole process, and that the total expenditure was US $0.40/m^3^.

Da et al. [26] prepared ZrO_2_ nanofiltration membranes by the sol–gel method. Cracks formed under the stress produced by the transformation from the tetragonal phase to the monoclinic phase in the heat treatment process of pure ZrO_2_ film, so they introduced glycerol to modify it. Experiments showed that glycerol modification played a positive role in the performance of the nanofiltration membrane. Glycerin inhibited the phase transition of ZrO2, and, with the increase in glycerol content, the pore diameter decreased, while the specific surface area increased. Taking high-salinity wastewater as the target pollutant, the performances of polymer nanofiltration membranes and ZrO_2_ nanofiltration membranes were compared. The experimental results showed that the flux of the ZrO_2_NF membrane was almost twice that of the polymerized nanofiltration membrane, and that the recovery rate was also superior to the other membranes. Lu et al. [27] used Zr(NO_3_)_4_, Ti(OC_4_H_9_)_4_, and C_3_H_8_O_3_ as precursors, and coated the prepared TiO_2_ on ZrO_2_ microporous materials. Doping or introducing glycerol [26] played a role in inhibiting the phase transition of ZrO_2_ and improved its stability. Satisfactory results were obtained when the materials were used in the treatment of simulated radioactive wastewater.

For the treatment of oil and gas wastewater, in addition to the membrane separation technology with single function, Riley [28] proposed a multi-barrier treatment scheme with the coupling of bioactive filtration and membrane technology (UF and NF), in order to alleviate the huge consumption of freshwater resources caused by drilling and hydraulic fracturing in the development of unconventional oil and gas resources. After pretreatment, the biodegradation of microorganisms in PW (the treatment of O&G produced water) can be used to remove the organic matter in oil and gas wastewater, which can effectively reduce the membrane pollution of subsequent membrane treatment and improve the treatment efficiency. Among them, the UF membrane adopted the polyamide flat NF membrane (NF90), and the NF membrane demonstrated two kinds of PVDF with different outer diameters. In particular, to avoid a difference in the results, the experimental water was obtained from different types of oil and gas produced water and backwater was fractured from three wells. The results showed that biological treatment was still an effective general technology, while UF and NF processes can deeply reduce turbidity and minimize the contribution of pollution. Finally, a large amount of dissolved solid and more than 99% organic matter can be removed from the wastewater.

In summary, nanomaterials can not only be used to make membrane materials, but can also be used to modify the membrane surface, so that both the transmittance and surface properties can be changed and enhanced, thus having a more superior separation performance. In addition, appropriate modification can effectively reduce membrane pollution and greatly improve economic benefits. The nano-modified membrane can be used to treat oil and gas wastewater, and can hopefully be widely used in food, medicine, cosmetics, and other fields.

## 3. Photocatalysis Technology

An advanced oxidation technology (AOPS) [29] is widely used in sewage treatment because of its low material price and high treatment efficiency for toxic pollutants. Table 3 summarizes some applications of TiO_2_ and nanocomposite in the photocatalytic treatment of oil and gas wastewater. TiO_2_ is often used in printing and dyeing wastewater because of its high photocatalytic activity and how it does not pollute to the environment. For the up-to-standard treatment of oil and gas wastewater, Bessa et al. [30,31] have proved that the semiconductor photocatalysis technology induced by TiO_2_ was a promising method in oil and gas wastewater treatment. Titanium dioxide had a large energy band gap which exceeded the REDOX potential of most organic compounds [32], and can effectively excite oxygen and OH· radicals in water, as shown in Figure 2. Finally, organic matter can be completely mineralized and transformed into a carbonaceous gas, which eliminates the need of subsequent biomass treatment [33]. As a result, it is widely used in induced photocatalysis.

### 3.1. Influence of Characteristics of Oil-Gas Wastewater on Photocatalysis

#### 3.1.1. Salinity

It is well known that high salt content is a typical characteristic of oil and gas wastewater [44]. Studies have shown that high concentration of Cl^−^ can significantly inhibit the degradation efficiency of photocatalysis [44], resulting in the inactivation of hydroxyl radicals(·OH). Mohammad et al. [45] proved that, in the photoreactor immobilized with TiO_2_, the existence of SO_4_^2−^ and CO_3_^2−^ can significantly reduce the degradation efficiency of p-toluene, ethylbenzene, and xylene.

#### 3.1.2. Organic Composition

Large amounts of drilling and fracturing of waste fluid will be produced in the process of oil and gas exploitation, including many additives, such as guanidine gum, formaldehyde, and other organic components [46]. Drilling wastewater also contains organic treatment agents such as potassium humate (KHm) and iron chromium lignosulfonate (FCLS) [47]. In a photocatalytic reaction, high concentrations of organic load will make the surface of TiO_2_ saturated, reducing photon efficiency, photonic efficiency, and its activity [48].

### 3.2. Influence of Photo-Catalysis Operating Conditions

In the process of photocatalytic degradation of oil and gas wastewater, in addition to improving the photocatalytic activity of the material, several factors affecting the photocatalytic process, which have been widely studied, are also briefly reviewed, such as catalyst concentration, pH, temperature, and oxidant.

#### 3.2.1. Catalyst Concentration

In the previous research works, Cheng et al. [49] have proved that 0.50 wt% Ag/TiO_2_ shows the strongest photocatalytic activity against POME. In order to obtain the optimal catalyst concentration, a series of concentration gradients (0.2, 0.5, 1.0, 1.5 and 2.0 g/L) were carried out with a photocatalytic reaction in a long-time range. The results showed that 1.0 g/L of Ag/TiO_2_ yielded the best photocatalytic degradation. The improvement of the efficiency was attributed to the production of more hydroxyl radicals (·OH) with an increase in the concentration, but, when the concentration was further increased, the photocatalyst was deactivated and the degradation efficiency was inhibited. In another report [50], it was also found that Pt/TiO_2_ showed lower activity when the concentration of photocatalyst exceeded a certain range. The reason was that the interaction between particles will be enhanced and that the light penetration will be affected when the concentration of catalyst exceeds the optimal dosage, resulting in an uneven light intensity distribution and a significantly reduced photocatalytic degradation efficiency [40].

#### 3.2.2. pH

When TiO_2_ was used to treat hydrocarbons in oily wastewater, it was found that the reaction was more favorable under strong acidic conditions. Wang [37] pointed out that pH had an important effect on the photoreaction process, not only on the surface charge distribution of TiO_2_ particles, but also on the hydroxyl radical (·OH). The increase in H^+^ concentration will be more conducive to the formation of the hydroxyl radical (·OH). Besides, TiO_2_ was positively charged on the oxide surface in acidic solution, thus, more negatively charged oils can be adsorbed [51]. Excessive acid or alkali can promote the agglomeration of particles and destroy the surface structure, so the solution should remain neutral [52].

#### 3.2.3. Temperature

In the treatment of refinery wastewater, Saien et al. [36] observed that, in most cases, the temperature increase can shorten the photocatalytic reaction time to achieve a higher COD removal rate, and that the higher the temperature, the faster the formation of TiO_2_ electronic holes. When the temperature reaches 318 K, the concentration of organic matter in the wastewater will change. Similar results can also be found in Ghasemi et al. [39]. However, when the reaction temperature was higher than 80 °C, the recombination of carriers and the desorption of the adsorbed reactants were enhanced, which led to a decrease in photocatalytic activity [48,53].

#### 3.2.4. Oxidant

O_2_ and H_2_O_2_ were generally selected as auxiliary catalysts for the photocatalytic reactions. When TiO_2_ and H_2_O_2_ coexist, more hydroxyl radicals (OH) were produced, and the initial reaction rate significantly improved [51]. Cazoir et al. [54] demonstrated that the combination of aeration and photocatalysis reduced the reaction time by half, compared with conventional photocatalysis. Wang et al. [55] put forward a set of comprehensive treatment technology for the treatment of fracturing backflow fluid in Henan Needle Oil Field, and achieved satisfactory results, that is, coagulation-oxidation-adsorption-photocatalysis, removal of some suspended solids (unbroken colloidal particles) in the early treatment stage, and further treatment of substances that made a great contribution to COD by oxidation and adsorption. Finally, fixed TiO_2_ was used as a photocatalyst for the advanced treatment of wastewater to achieve the reinjection mark.

### 3.3. Limitation of Nano-TiO_2_ and Enhancement of Its Photocatalysis Activity

The limitation of TiO_2_-induced photocatalysis in oil and gas wastewater treatment is mainly reflected in the following two aspects:

#### 3.3.1. Difficult to Recycle

Nano-TiO_2_ has great potential in the photocatalytic treatment of oil and gas wastewater. However, the recovery method of nano-TiO_2_ particles has not made great progress, which greatly limits the application of photocatalysis technology in oil and gas wastewater treatment. Mascolo et al. [56] tried to fix the catalyst on the inner wall of the reactor, but it would lead to a decrease in the specific surface area of TiO_2_ particles, while Chen et al. [57] prepared N-doped TiO_2_/diatomite composites using pretreated diatomite as a carrier and achieved good results in the treatment of printing and dyeing wastewater. Diatomite not only restrained the aggregation of nano-TiO_2_ particles but also significantly improved the recyclability of the materials. Similar materials include kaolinite [58] and graphene [41]. Zielinska-Jurek et al. [43] proved that Fe_3_O_4_@SiO_2_/TiO_2__P25 was easier to separate and recover than other photocatalytic materials because of its magnetism, and that the ratio of Fe_3_O_4_ to TiO_2_ was 2:1.

#### 3.3.2. Narrow Light Absorption Range

Because of the high band gap (3.2 eV) of TiO_2_, it can only absorb UV light which accounts for only 3% and 5% of the full spectrum. Common modifications include doping of metal/non-metal ions (Ag, Pt, V, C, N, F), precious metal deposition, semiconductor recombination, and so on. Ren et al. [59] prepared C-doped TiO_2_ by using a simple hydrothermal synthesis method. Under visible light irradiation, C-doped TiO_2_ showed higher photocatalytic activity than TiO_2_ and P25 in the degradation of RhB. The degradation rate of 1 wt% F-diatomite/TiO_2_ to RhB reached 90% [60]. Shao et al. [61] prepared a V-TiO_2_ photocatalyst with good photocatalytic performance under natural light irradiation by an improved sol–gel method. Cheng et al. [49] synthesized a composite catalyst with visible light activity by reacting silver nitrate with titanium dioxide powder. The results showed that the addition of Ag successfully narrowed the band gap from 3.20 eV to 2.50 eV of pure TiO_2_ photocatalyst.

In this section, the influencing factors of TiO_2_-induced photocatalytic treatment in the application of oil and gas wastewater and the limitations of its applicability are reviewed. Nevertheless, this method is still widely used for its high efficiency, and its green and economic performance. TiO2-induced photocatalytic treatment is still worthy of further study, and its limitations are expected to be improved in the future.

## 4. Conclusions and Outlook

This review summarizes the progress of nanomaterials in the treatment of oil and gas wastewater. Although there are diverse methods for the treatment of oil and gas wastewater, most of them are still in the stage of experimental research, and a large-scale application of a win-win treatment method with economic and environmental benefits has not been found yet.

Nano-TiO_2_ semiconductor photocatalysis technology was proved to be attractive in wastewater treatment because of its strong photocatalytic activity, lack of secondary pollution, and low selectivity, while the complex inorganic and organic substances and high energy consumption of oil and gas wastewater will restrain the effect of photocatalysis to some extent. The main goal is to improve the catalyst activity and reduce the cost through modification. In addition, the cleaning and replacement requirements of various membrane materials used in membrane separation technology do not keep up with the development of separation technology. Some membrane materials need chemical cleaning, which will lead to secondary pollution, an important factor that restricts the industrial application of membrane separation technology.

The separation membrane with a single function has some problems, such as poor separation efficiency and a difficulty to remove trace pollutants, while the single photocatalysis technology has the problem of catalyst recovery. The preparation of the photocatalytic separation membrane by the combination of photocatalysis technology and membrane separation technology cannot only improve the separation and decomposition efficiency of organic matter, but also realize the reuse of the membrane, reduce the cost, and reduce the pollution in the environment. 

## Figures and Tables

**Figure 1 molecules-26-03945-f001:**
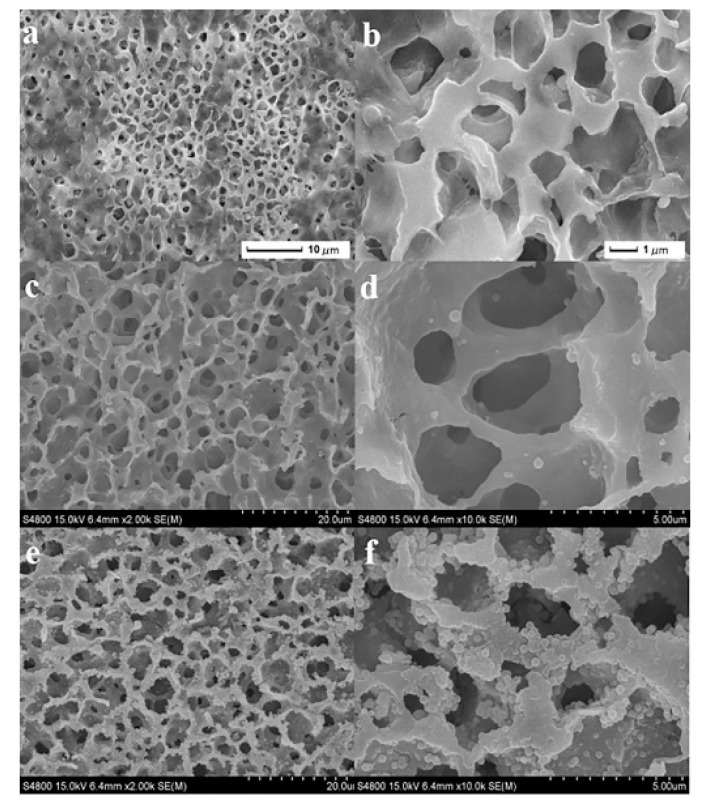
SEM images of (**a**,**b**) PVDF; (**c**,**d**) PVDF@pDA; and (**e**,**f**) PVDF@pDA@SiO_2_ membranes [21].

**Figure 2 molecules-26-03945-f002:**
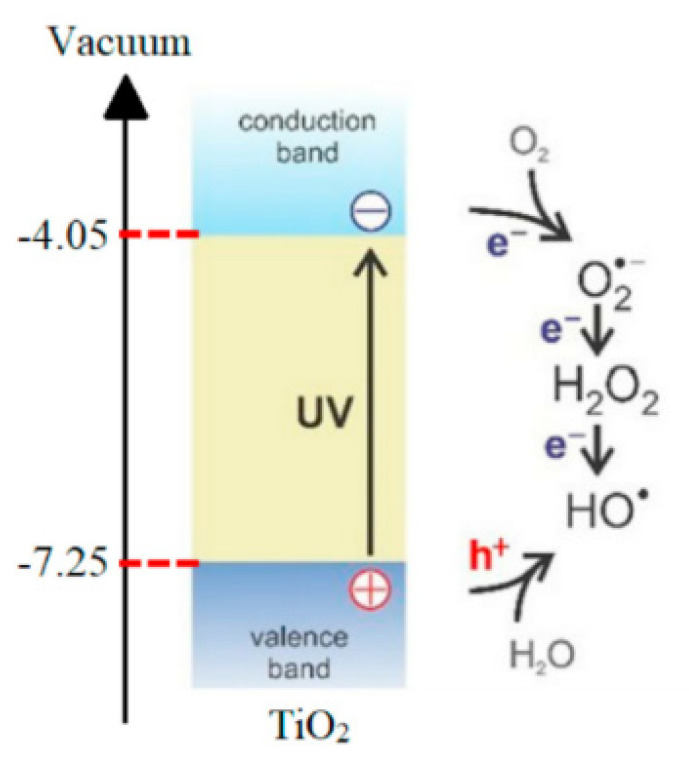
Band edges of TiO_2_ and generation of OH· radicals from O_2_ and water molecules upon UV-excitation [34].

**Table 1 molecules-26-03945-t001:** The classification of membrane separation technology.

Separation Membrane	Pore Size	Application
MF	0.1~10 μm	Filter out most suspended solids, bacteria and other impurities. Viruses and ions can pass through
UF	2~100 nm	Filter out most proteins and other macromolecules and viruses
NF	0.5~1 nm	Filter out divalent and multivalent ions, such as SO_4_^2−^, Ca^2+^, Mg^2+^
RO		Only water can pass through

**Table 2 molecules-26-03945-t002:** Nano-material modified membrane.

Nano-Materials	Preparation Method of Modified Membrane	Performance	Ref.
Tubular nanofibers and micro/nano spheres	Electrostatic spraying	3D hierarchical micro/nano structure; Super hydrophilic; The morphology of the covered micro/nano spheres can be easily controlled; High separation efficiency	[14]
nano-Al_2_O_3_	Impregnation method	The oil repellent and hydrophilic properties are enhanced; With an increase in surface roughness, the membrane fouling decreases	[15]
nano-ZnO	Chemical deposition	It exhibits good separation performance for highly corrosive aqueous solutions and light oil/heavy oil mixtures	[16]
nano-TiO_2_	Impregnation method	The separation rate of oil-water emulsion is high. It shows excellent anti-pollution performance and recyclability	[17]
nano-Ag	In situ co-mixed reduction method	It can effectively reduce and separate macromolecular pollutants	[18]
nano-SiO_2_	Thermally induced phase separation	The modified film has super hydrophilic and superhydrophobic properties; It has high oil-water separation efficiency	[19]

**Table 3 molecules-26-03945-t003:** A summary of photocatalysis for different types of oil and gas wastewater.

Wastewater Type	Catalyst	Light Source	Major Funding	Ref.
Bilge water	TiO_2_/KOH	370W UV lamp	The oil is completely decomposed	[30]
Oil produce water	P25	125 UV lamp	DOC removal 90% in 7 days	[35]
Refinery wastewater	P25	400W UV lamp	COD removal 90% in 240 min	[36]
Fracturing wastewater	Bentonite loading TiO_2_-Ag_2_O	Visible light	COD removal 58.1% 180 min	[37]
Diesel-polluted surface water	Silver/titanium dioxide/graphene ternary nanoparticles	Visible light (500-W halogen tungsten lamp with a UV cutoff filter)	Diesel oil removal efficiency was 75% in 16 h	[38]
Petroleum refinery wastewater (PRWW)	TiO_2_/Fe-ZSM-5	8W UV lamp	COD removal 80% in 240 min	[39]
Petroleum refinery wastewater	P25	400W UV lamp	TCOD removal 83% in 120 min	[40]
Diesel oil	Ni-N-TiO_2_/PEGC	Visible light (500W xenon lamp with the UV cutoff filter)	Diesel oil removal efficiency was 95.9% in 5 h	[41]
Weathered oil	Food-grade TiO_2_	4W UV lamp	DOC increase 60% in 24 h	[42]
Fracturing wastewater	Fe_3_O_4_@SiO_2_/TiO_2__P25	UV	COD removal 40% in180 min	[43]
Oil and gas produce water	ZnO_2_	Visible light (a solar simulator fitted with IR filter)	TOC removal 20% in 7 h	[29]

## Data Availability

We haven’t report any data in this review paper.

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
