# Peer review of "The Perspective and Challenge of Nanomaterials in Oil and Gas Wastewater Treatment"

_molecules, 2021, doi:10.3390/molecules26133945_

Round 1

Reviewer 1 Report

The manuscript deals with an interesting topic of oil and gas wastewater treatment using nanomaterials. It introduced and summarised the use of membranes and photocatalysis, but there is a lack of discussion. This current version should be further developed as a review article to be suitable for publication in Molecules.

Specific comments:

The target wastewater in the review is produced from oil and gas extraction industry. What types of components the wastewater contain depending on various sources, fields, or location? Is there any regulation to be restricted to discharge the wastewater?

The authors misuse abbreviation terms in most cases. One the abbreviation is defined, it does not have to repeat. For example, after the term “nanofiltration (NF)” is mentioned in introduction section, only NF is enough in the rest of manuscript.

Why are membranes regarded as nano materials? Usually they are classified as polymers, inorganics and their composites. Some nanocomposite membranes, e.g. polymers with nano-sized fillers, can be regarded as nano materials, but the authors did not mention it. The section 2 should be more carefully reviewed and rearranged with proper categorised sub-sections. It can be divided to MF, UF, NF and RO, as suggested in Table 1, or polymers, inorganics and nanocomposites depending on membrane materials. In the current version, the authors just listed the reported works with brief summary in few sentences, that the review paper should include in-depth discussion of each papers and research trends.

Similar in the section 3; the authors mentioned and listed properties and influence on photocatalysis, there is a lack of discussion why the results was obtained.

Table 1 – MF membranes have pore sizes in microns (0.1-10 um), not meter.

“Chen et al. [16] prepared a sandwich-like nanofiltration membrane of PA6@GO@PA6, by intercalating hydrophilic nanoparticles such as carbon dioxide, silicon dioxide and trisilicon tetroxide with substrate” – carbon dioxide is Titanium dioxide in [16] as I checked in the reference.

Bivalent in page 2 is to be divalent.

The practical application for the wastewater treatment is essential. Is there any pilot or commercialised plants for these wastewater treatment using nanomaterials?

A brief summary of the take-away points at the end of subtopic 2 and 3 can help readers grasp the overall picture of the current progress in different aspects of oil and gas wastewater treatment technology. 

Author Response

Comments:

  • The target wastewater in the review is produced from oil and gas extraction industry. What types of components the wastewater contain depending on various sources, fields, or location? Is there any regulation to be restricted to discharge the wastewater?

Response:

  • Thanks for your comment. As mentioned in the introduction, oil and gas wastewater mainly contains complex organic compounds such as polycyclic aromatic hydrocarbons, aromatic hydrocarbons, oil, heavy metals and natural radionuclides. The discharge of wastewater needs to remove harmful substances such as heavy metals and complex organic compounds such as oil.
  • The authors misuse abbreviation terms in most cases. One the abbreviation is defined, it does not have to repeat. For example, after the term “nanofiltration (NF)” is mentioned in introduction section, only NF is enough in the rest of manuscript.

Response:

  • We are very sorry for our negligence. According to this comment, we have carefully revised all the use of abbreviation terms.
  • Why are membranes regarded as nano materials? Usually they are classified as polymers, inorganics and their composites. Some nanocomposite membranes, e.g. polymers with nano-sized fillers, can be regarded as nano materials, but the authors did not mention it. The section 2 should be more carefully reviewed and rearranged with proper categorised sub-sections. It can be divided to MF, UF, NF and RO, as suggested in Table 1, or polymers, inorganics and nanocomposites depending on membrane materials. In the current version, the authors just listed the reported works with brief summary in few sentences, that the review paper should include in-depth discussion of each papers and research trends.

Response:

  • Thanks for your comment. Nano materials can be used to modify the surface of the membrane, so that the membrane has some characteristics of nano materials, such as micro pore size or structural stability. We have carefully rearranged with proper categorised sub-sectionsof the manuscript according to the reviewers' comments.

(4) Similar in the section 3; the authors mentioned and listed properties and influence on photocatalysis, there is a lack of discussion why the results was obtained.

Response:

  • Thanks for your useful suggetion. We have carefully revised the manuscript according to the reviewers' comments.The content of the review is supplemented according to the following literature:

[33] Kim Hoong Ng. Adoption of TiO2-photocatalysis for palm oil mill effluent (POME) treatment: Strengths, weaknesses, opportunities, threats (SWOT) and its practicality against traditional treatment in Malaysia[J]. Chemosphere, 2021.

[34] Labuz, P. et al. Photo-generation of reactive oxygen species over ultrafine TiO2 particles functionalized with rutineligand induced sensitization and crystallization effect[J]. Res Chem, 2019.

(5) Table 1 – MF membranes have pore sizes in microns (0.1-10 um), not meter. 

Response:

  • Thanks for your useful sugggetion. We have carefully revised the manuscript according to the reviewers' comments.

(6) Chen et al. [16] prepared a sandwich-like nanofiltration membrane of PA6@GO@PA6, by intercalating hydrophilic nanoparticles such as carbon dioxide, silicon dioxide and trisilicon tetroxide with substrate” – carbon dioxide is Titanium dioxide in [16] as I checked in the reference.

Response:

  • We are very sorry for our negligence. According to this comment, we have carefully proofread and modify the contents of the reference cited.

(7) Bivalent in page 2 is to be divalent.

Response:

  • We are very sorry for our negligence. We have revised the term according to your comments.

(8) The practical application for the wastewater treatment is essential. Is there any pilot or commercialised plants for these wastewater treatment using nanomaterials?

Response:

  • Thanks for your comment.The research on nano material modified membrane for wastewater treatment is more extensive, and it has been applied and practiced in real life. 

(9) A brief summary of the take-away points at the end of subtopic 2 and 3 can help readers grasp the overall picture of the current progress in different aspects of oil and gas wastewater treatment technology.

Response:

  • Thanks for your comment.We have revised and supplemented the review according to your comments.

Reviewer 2 Report

In this study, the authors reviewed the roles of nanomaterials in improving oil/gas wastewater treatment by integrating them into membrane matrix and performing as photocatalysts. The manuscript has been well organized and written. The major concern is that there is lack of perspective in each section. In addition, the reviewer also has several comments, which need to be further responded.

  • Introduction, last paragraph: It should change to “(1) integrating with membrane separation technology……”
  • It suggests moving the first two-paragraph in Section 2. Membrane separation technology to the Introduction section (before the last paragraph). Then the title of Section 2 should be changed to “Modification of membranes with nanomaterials”.
  • It suggests adding a Table summarizing the modification condition (such as fabrication process, doses….) and performance of nanomaterial-modified membranes during oil/gas wastewater treatment.
  • In the Section 2, the authors should also give more details on how the modified membranes improved the treatment performance and why these modified membranes had better performance.
  • Moving “Table 2 summarizes nanocomposites in photocatalytic treatment of oil and gas wastewater” after “toxic pollutants”.
  • Title 3.2: Influence of photo-catalysis operating conditions
  • As several research works on integrating membrane separation and photo-catalysts have been published, it suggests adding a section to review the combined technique (instead of only considering as an “Outlook” point).
  • When the information in the literatures was described, the “past tense” should be used.

Author Response

Comments:

  • Introduction, last paragraph: It should change to “(1) integrating with membrane separation technology……”

Response:

  • Thanks for your comment.We have adjusted the content distribution according to your suggestion.
  • It suggests moving the first two-paragraph in Section 2. Membrane separation technology to the Introduction section (before the last paragraph). Then the title of Section 2 should be changed to Modification of membranes with nanomaterials”.

Response:

  • We agree. Thanks for your comment.We have adjusted the content distribution according to your suggestion.
  • It suggests adding a Table summarizing the modification condition (such as fabrication process, doses….) and performance of nanomaterial-modified membranes during oil/gas wastewater treatment.

Response:

  • Thanks for your useful suggestion. We have added a summary form as you suggested.

Nano-materials

Preparation method of modified membrane

Performance

Ref.

Tubular nanofibers and micro/nano spheres

Electrostatic spraying

3D hierarchical micro/nano structure; Super hydrophilic; The morphology of the covered micro/nano spheres can be easily controlled; High separation efficiency

[14]

nano-Al2O3

Impregnation method

The oil repellent and hydrophilic properties are enhanced; With the increase of surface roughness, the membrane fouling decreases

[15]

nano-ZnO

Chemical deposition

It exhibits good separation performance for highly corrosive aqueous solutions and light oil/heavy oil mixtures

[16]

nano-TiO2 

Impregnation method

The separation rate of oil-water emulsion is high. It shows excellent anti-pollution performance and recyclability

[17]

nano-Ag

In situ co-mixed reduction method

It can effectively reduce and separate macromolecular pollutants

[18]

nano-SiO2

Thermally induced phase separation

The modified film has super hydrophilic and superhydrophobic properties; It has high oil-water separation efficiency

[19]

[14] T. Zhang et al., One-step preparation of tubular nanofibers and micro/nanospheres covered membrane with 3D micro/nano structure for highly efficient emulsified oil/water separation[J],Journal of the Taiwan Institute of Chemical Engineers, 2021.

[15] Yusuf Olabode Raji, Mohd Hafiz Dzarfan Othman, Nik Abdul Hadi Sapiaa Md Nordin et al. Wettability Improvement of Ceramic Membrane byIntercalating Nano-Al2O3 for Oil and Water Separation[J], Surfaces and Interfaces, 2021.

[16] Elmira Velayi, Reza Norouzbeigi. A mesh membrane coated with dual-scale superhydrophobic nano zincoxide: Efficient oil-water separation[J]. Surface & Coatings Technology, 2020.

[17] Ze Li, Zhen-Liang Xu. Three-channel stainless steel hollow fiber membrane with inner layer modified by nano-TiO2coating method for the separation of oil-in-water emulsions[J]. Separation and Purification Technology, 2019.

[18] Xiaofeng Fanga, Jiansheng Lia, Bangxing RenPolymeric. ultrafiltration membrane with in situ formed nano-silver within the inner pores for simultaneous separation and catalysis[J]. Journal of Membrane Science, 2019.

[19] Jian Pan, Changfa Xiao. ECTFE hybrid porous membrane with hierarchical micro/nano-structural surface for efficient oil/water separation[J]. Journal of Membrane Science, 2017.

(4) In the Section 2, the authors should also give more details on how the modified membranes

improved the treatment performance and why these modified membranes had better performance.

Response:

  • Thanks for your comment.We have adjusted the content distribution according to your suggestion.

(5) Moving “T able 2 summarizes nanocomposites in photocatalytic treatment of oil and gas

wastewater” after “toxic pollutants”.

Response:

  • Thanks for your comment.We have adjusted the content distribution according to your suggestion.

(6) Title 3.2: Influence of photo-catalysis operating conditions

Response:

  • Thanks for your comment.We have adjusted the content distribution according to your suggestion.

(7) As several research works on integrating membrane separation and photo-catalysts have been published, it suggests adding a section to review the combined technique (instead of only considering as an “Outlook” point).

  • Thanks for your comment. The combination of membrane separation and photocatalysis has been less studied, and it is expected to have more mature and extensive research and application.

(8) When the information in the literatures was described, the “past tense” should be used.

Response:

  • Thanks for your comment.We have modified the tense problem of the reference section.

Round 2

Reviewer 1 Report

The revised manuscript is improved well.